# Brain activations associated with anticipation and delivery of monetary reward: A systematic review and meta-analysis of fMRI studies

S. Jauhar[1], L. Fortea[2], A. Solanes[2,3,4], A. Albajes-Eizagirre[2,3,5], P. J. McKenna[2,5]*, J. Radua[1,2,3,5,6]

1 Department of Psychosis Studies, Institute of Psychiatry, Psychology and Neuroscience, King's College London, United Kingdom, 2 Imaging of Mood- and Anxiety-Related Disorders (IMARD) group, Institut d'Investigacions Biomèdiques August Pi i Sunyer (IDIBAPS), Barcelona, Spain, 3 FIDMAG, Germanes Hospitalàries Research foundation, Barcelona, Spain, 4 Antonomous University of Barcelona, Barcelona, Spain, 5 Centro de Investigación Biomédica en Red de Salud Mental (CIBERSAM), Madrid, Spain, 6 Center for Psychiatry Research, Department of Clinical Neuroscience, Karolinska Institute, Stockholm, Sweden

* mckennapeter1@gmail.com

**Data Availability Statement:** The relevant dataset (peak activation co-oridinates for the each of the individual studies included in the meta-analysis)

## Abstract

### Background

While multiple studies have examined the brain functional correlates of reward, meta-analyses have either focused on studies using the monetary incentive delay (MID) task, or have adopted a broad strategy, combining data from studies using both monetary and non-monetary reward, as probed using a wide range of tasks.

### Objective

To meta-analyze fMRI studies that used monetary reward and in which there was a definable cue-reward contingency. Studies were limited to those using monetary reward in order to avoid potential heterogeneity from use of other rewards, especially social rewards. Studies using gambling or delay discounting tasks were excluded on the grounds that reward anticipation is not easily quantifiable.

### Study eligibility

English-language fMRI studies (i) that reported fMRI findings on healthy adults; (ii) that used monetary reward; and (iii) in which a cue that was predictive of reward was compared to a no win (or lesser win) condition. Only voxel-based studies were included; those where brain coverage was incomplete were excluded.

### Data sources

Ovid, Medline and PsycInfo, from 2000 to 2020, plus checking of review articles and meta-analyses.

has been uploaded to FigShare. It can be found at Title: Individual reward studies anticipation and delivery coordinates for meta-analysis (Jauhar et al, 2021), Author Peter McKenna; https://figshare.com/articles/dataset/Individual_reward_studies_anticipation_and_delivery_coordinates_for_meta-analysis_Jauhar_et_al_2021_/15022347.

**Funding:** This work was supported by a Miguel Servet II Research Contract CPII19/00009 and Research Project PI19/00394 from the Plan Nacional de I+D+i 2013–2016, the Instituto de Salud Carlos III-Subdirección General de Evaluación y Fomento de la Investigación and the European Regional Development Fund (FEDER). The funders had no role in study design, data collection and analysis, decision to publish, or preparation of the manuscript.

**Competing interests:** The authors have declared that no competing interest exist.

### Data synthesis

Data were pooled using Seed-based d Mapping with Permutation of Subject Images (SDM-PSI). Heterogeneity among studies was examined using the $I^2$ statistic. Publication bias was examined using funnel plots and statistical examination of asymmetries. Moderator variables including whether the task was pre-learnt, sex distribution, amount of money won and width of smoothing kernel were examined.

### Results

Pooled data from 45 studies of reward anticipation revealed activations in the ventral striatum, the middle cingulate cortex/supplementary motor area and the insula. Pooled data from 28 studies of reward delivery again revealed ventral striatal activation, plus cortical activations in the anterior and posterior cingulate cortex. There was relatively little evidence of publication bias. Among moderating variables, only whether the task was pre-learnt exerted an influence.

### Conclusions

According to this meta-analysis monetary reward anticipation and delivery both activate the ventral but not the dorsal striatum, and are associated with different patterns of cortical activation.

## Introduction

Recognition of the existence of a 'reward circuit' in the brain, ie a system that mediates the effects of positive reinforcement on behaviour, dates back to Olds and Milner's [1] discovery of the rewarding properties of electrical brain stimulation. Subsequently it was shown that catecholamines were involved in the effect [2], and then that dopamine rather than noradrenalin was the important transmitter (see [3]). A recent important development has been Schultz's [4] demonstration that around 75% of midbrain dopamine neurons–those that give rise to the ascending 'mesotelencephalic' projections [5]–show a switch from tonic activity to phasic bursts consistent with their coding a reward prediction error signal [6].

These findings do not in themselves permit the reward system to be anatomically defined, not least because the mesotelencephalic dopamine system projects widely and in man innervates the entire cerebral cortex [5]. Some further evidence on this question comes from animal studies, where single cell recording studies have established two main areas where firing patterns change during reward processing. One of these is the striatum, including the caudate nucleus and putamen, particularly their ventral sectors [7, 8]. The other is the orbitofrontal cortex, where neurons that respond to various aspects of reward have been found [7, 9]. Neurons showing reward responsiveness have, however, also been documented in a number of other brain regions, including the amygdala [10], the dorsolateral prefrontal cortex [11, 12], the anterior cingulate cortex [13], the subthalamic nucleus, the substantia nigra pars reticulata, the pallidum, the lateral hypothalamus, the nucleus basalis of Meynert, the parietal cortex and the premotor cortex [11, 13, 14].

The other main source of evidence on the anatomical localization of the reward system comes from human functional imaging studies. Early studies using PET and fMRI found

activations in a number of brain regions in response to stimuli ranging from seeing attractive faces, to viewing erotic videos, and to being administered doses of cocaine [15] (note: throughout this article we use the term 'activations' for convenience, although strictly speaking what is being described are increases in blood flow/metabolism in PET and increases in BOLD signal in fMRI). While PET findings have been heterogeneous [16–22], from the outset studies employing fMRI found a more consistent pattern involving particularly the nucleus accumbens (part of the ventral striatum), the amygdala and the orbitofrontal cortex [15].

Further work in this area has depended heavily on the development of the monetary incentive delay (MID) fMRI paradigm by Knutson and co-workers [23–25]. In this, subjects are trained on a reaction time task (eg pressing a button when they see a white square), whose difficulty is individually adjusted during a training phase so that they are successful approximately two-thirds of the time. The subjects then perform the same task in the scanner, with the reaction time stimulus (the white square) being preceded by a cue, (eg a circle) which signals that performing the task successfully will lead to a certain amount of money being won, or by a different cue (eg a triangle), which indicates that successful performance will not be followed by reward. Feedback about whether money has been won is given visually immediately after the response is made. Event-related fMRI is used to measure activations at the time of perceiving the reward predictive cue (reward anticipation or expectation) and at the moment of receiving feedback that reward has been received (reward delivery, also referred to as outcome or receipt). In some versions of the task additional information (eg horizontal lines superimposed on the cues) indicates that different amounts of money will be won on different trials. Another common variant uses a second cue which indicates that subjects will lose money if they do not successfully perform the reaction time task.

In an initial meta-analysis of studies using the MID, Knutson and Greer [26] pooled data from 21 voxel-based fMRI studies and found evidence for activation associated with reward anticipation in the medial frontal gyrus, the nucleus accumbens, the anterior insula, the putamen and the thalamus. In this meta-analysis, it should be noted, reward anticipation was not compared against a no win baseline, but instead against anticipation of loss. More recently, Oldham et al [27] meta-analyzed 49 voxel-based MID studies and found that reward anticipation was associated with activation in a large cluster encompassing both ventral and dorsal sectors of the striatum, the thalamus and amygdala, all bilaterally, and the midbrain. Cortical regions activated included the insula, the premotor cortex and supplementary motor area, the occipital cortex and the cuneus; differently to in Knutson and Greer's [26] meta-analysis, clusters in the medial frontal cortex were not seen. Reward delivery was associated with activation in the ventral striatum and the amygdala in 27 studies, with clusters of cortical activation in the medial frontal and orbitofrontal cortex and in the posterior cingulate cortex. Wilson et al [28] had broadly similar findings for reward anticipation in a meta-analysis of 15 MID studies that utilized information from group maps provided by authors rather than peak co-ordinates; in this meta-analysis the cortical regions activated were more extensive than in Oldham et al's [27] meta-analysis.

Other meta-analyses have opted for a broader strategy, pooling data from studies using a wide range of paradigms involving anticipation and/or delivery of monetary as well as other kinds of reward, such as juice, water, others, a smiling face, or simply knowledge of having performed the task correctly [29–31]. Clearly, broadening the range of different rewards beyond just money means that if these produce different patterns of activation, the results will be conflated. This is arguably of relatively little concern when the other rewards have innate reinforcing qualities, such as food and drink, but it is not clear that this assumption holds true when the reward is social in nature. These meta-analyses all also included studies where there was no cue-reward contingency–ie there was no explicit contrast between a cue that predicts reward

and one that predicts no reward (or less reward)–for example, gambling and so-called delay discounting tasks, where subjects have to choose between a small certain or a larger uncertain reward. Such studies do not include one condition where a cue signals that money can be won and another where no money (or a smaller amount of money) will follow, meaning that extraction of findings for reward anticipation is either not possible or has to be inferred indirectly.

A further consideration when pooling data from voxel-based reward studies is whether the whole brain is covered in the examination. Studies not covering a region will fail to find and report potential activations in that region, biasing the effect size for partially-covered brain regions towards zero. This problem applies to some, though not all studies carried out up to around 2012, after which time whole brain coverage appears to have become standard. The issue has not been addressed in existing meta-analyses.

The present meta-analysis attempted to steer a middle course between the narrow and broad meta-analytic approaches taken so far in the literature. Specifically, we restricted the analysis to studies using monetary reward in order to avoid the potential confounding factor that non-monetary (especially social) rewards might activate different brain areas than monetary reward. We also only included studies in which there was an overt monetary cue-reward contingency, ie where one cue signalled that money could be won and another that indicated that no or a lesser amount of money could be won. Finally, we excluded studies that did not employ whole brain coverage. We carried out separate meta-analyses of studies examining activations at the time of reward anticipation, ie at the time of presentation of a cue that predicted monetary reward, and of reward delivery, ie at the time of subsequent receipt of the reward.

Data were pooled using Seed-Based d Mapping (SDM) [32, 33] with permutation of subject images (SDM-PSI) [34–36] which has a number of advantages over other voxel-based meta-analytic methods [32]. Specifically, previous methods have relied on an unorthodox statistical test: they aim to find those voxels whether the convergence of findings is statistically significantly higher than in other voxels. Conversely, the statistical tests in SDM-PSI aim to find those voxels whether the BOLD response is statistically significant, ie as in the standard statistical tests applied to original imaging data within SPM or FSL. The use of standard statistics has multiple additional benefits, such as taking the effect size into account (other methods do not), reporting standard estimates of between-study heterogeneity (e.g., the $I^2$ statistic, interpreted as the percentage of variation unrelated to sampling error), and allowing standard tests for the detection of potential publication bias based on funnel plot asymmetry (for more details see [32]).

## Method

Papers were searched in Ovid, Medline and PsycInfo, from 2000, the year the first paper using the MID was published, to April 17th 2020. Search terms were (("fMRI" AND ("reward" OR "prediction error" OR "reinforcement learning" OR "monetary incentive delay task")). Limits included abstracts and humans and yr = "2000-Current"). We also checked the reference lists of all papers obtained plus review articles and meta-analyses. When studies reported on overlapping samples of participants, the larger sample was used if it provided usable data. There was no protocol for the meta-analysis. For a PRISMA checklist see S1 Checklist.

Only published studies reported in English were included. We included voxel-based fMRI studies that (i) reported fMRI findings on healthy adults (including the healthy control groups of clinical studies and the placebo groups of drug studies); (ii) used monetary reward; and (iii) in which there was an overt cue-reward contingency and a comparison no win (or lesser win) condition. With respect to (i), studies were excluded if the samples consisted entirely of elderly

(age>65) or adolescent samples (age<18), although studies where only some of the participants were over 65 or under 18 were included. With respect to (ii), we excluded studies where the participants did not actually receive money at the end of the study, although we accepted studies where they were only given a proportion of their winnings, or they received a fixed amount of money. With respect to (iii), we did not include studies where the participants simply had to choose among different strategies for winning money (ie gambling and delay discounting studies, see above). Otherwise, we defined the concept of cue broadly: it could include an arbitrary stimulus, such as a shape, a complex design or an indoor or outdoor scene. Studies were also included where the cue had an innate, non-arbitrary relationship to the reward, eg a pie chart indicating the probability of winning, or the paradigm featured of one of a number of visually distinct 'slot machines', or simply a visual representation of the amount that could be won on the trial; this was on the basis that in these cases a valid cue-reward predictive relationship existed even though it was obvious rather than had to be deduced by trial and error.

Studies could report fMRI findings at any level of statistical significance (SDM only uses the information about peak activations to recreate a map of effect size); however, studies using small volume correction were excluded. Studies reporting findings from multiple ROIs were also excluded. As noted above, we required that the whole brain be covered in the scan; this was operationally defined as axial slices encompassing least 9.6 cm or coronal slices encompassing a minimum of 14.5cm, or alternatively an explicit statement that there was whole brain coverage.

For the meta-analysis of reward anticipation, we considered analyses where reward predictive cue activation was measured compared to a neutral, non-reward-predicting cue, or the comparison was between a high value and a low value cue, or where activation was measured as a linear correlation with cues signalling different amounts of monetary reward. However, we did not include analyses where the comparison was between a reward-predicting cue and a loss-predicting cue or where the linear correlation included monetary loss; this was because of uncertainty about whether the response to loss predicting cues would engage only the reward system and not, say, a punishment-related system. Studies where the comparison was not between monetary predictive and non-predictive cues were also excluded, eg where the comparison was between monetary reward and verbal reward.

For the meta-analysis of reward delivery, studies were included which compared activations at the time feedback was given and indicated that the subject had won or not won money. In terms of the no-win baseline condition we counted failure to receive reward both because of failure to perform the interpolated task sufficiently well, or because the preceding cue indicated no reward would be won. As in the reward anticipation meta-analysis, studies that compared reward delivery against loss delivery were not considered.

Data extraction was independently conducted by two members of the team (PJM and SJ), with fMRI methodological issues and discrepancies being discussed with a third (JR). In cases where a relevant analysis was carried out, but the peak activation co-ordinates in MNI or Talairach space were not given (eg because not all the co-ordinates were reported or the results were only shown in a figure), authors were contacted.

For data pooling using Seed-Based d Mapping (SDM) [32, 33] with permutation of subject images (SDM-PSI) [34–36], coordinates were first converted to a common MNI space using the Lancaster method (taking into account the small changes in MNI space between SPM and FSL, and undoing the MNI conversions conducted with the old Brett method) [37]. Second, the map of the potential lower and upper bounds of possible effect sizes (Hedges' g) was created for each study based on the level of statistical significance, the coordinates and effect size of the reported peaks, and the anisotropic covariance between adjacent voxels [33]. Third,

multiple effect size maps (and the corresponding variance maps) were imputed voxelwise for each study, adding normal spatially correlated noise to the maximum likely effect sizes [35]. Fourth, images of each dataset of imputed effect size maps were combined using a standard random-effects meta-analysis, and the meta-analytic maps resulting from the different imputations were combined using Rubin's rules in a single Hedges' g map and a single variance map. Finally, subject images were imputed for each study and statistical significance was assessed via permutation test of the subject images [35]. In the text, we consider the most robust results (FWER < 0.01 in clusters of at least 100 voxels); for completeness, we also report results at a more liberal threshold (FWER < 0.05) with clusters of at least 10 voxels in the (see S1 File). All results are reported in MNI space and with brain regions labelled according to the AAL atlas.

We conducted tests to evaluate the robustness of the main findings. For each cluster that emerged we examined heterogeneity in the peak using the $I^2$ statistic. While the use of $I^2$ as an indicator of heterogeneity is not without problems [38], researchers usually consider $I^2 > 50\%$ as an indicator of significant heterogeneity. Potential publication bias was examined for the peak of each emergent cluster using funnel plots. These were visually inspected to find asymmetries in which small studies reported larger effect sizes than large studies, which could indicate that some small studies with weak of null effect sizes had not been published. We also formally tested these asymmetries conducting meta-regressions by standard error, conceptually similar to the Egger test [36].

A number of potential moderator variables were examined using meta-regression. These were: whether or not the task was pre-learnt outside the scanner, percentage of males in the sample, the amount of money won on successful trials, and the full width at half maximum (FWHM) of the smoothing kernel used. This last variable was included because Sacchet and Knutson [39] found that peak activation co-ordinates in studies that used smaller spatial smoothing kernels (i.e. <6 mm FWHM) were more anterior than those identified for studies that used larger kernels (i.e. >7 mm FWHM). For reward anticipation, the probability of reward–ie the probabilistic frequency with which reward would later be delivered on viewing the anticipation cue–was also examined.

## Results

A flow diagram of the study selection process is shown in Fig 1. Forty-nine studies were included. Forty-seven studies provided peak activation co-ordinates for reward anticipation, of which two [40, 41] could not be included because accompanying t- or z-score information was not available. Twenty-nine studies of reward delivery were found, with one [41] again not being included due to non-availability of t- or z-scores.

Details of the studies included in the two meta-analyses are shown in Table 1. More details are available from the authors on request. Excluded studies, with reasons, are listed in the (see S1 File).

### Reward anticipation

The findings are shown in Fig 2 and Table 2. The analysis revealed nine clusters of statistically significant voxels. The largest (5644 voxels, peak at MNI 16,6,-12 with Hedges' g = 0.42) covered areas of the basal ganglia, particularly the ventral striatum extending bilaterally into the insula. It also reached small areas of the superior temporal cortex, the inferior frontal cortex bilaterally, the olfactory cortex, and the gyrus rectus, all bilaterally, and to a very small extent the amygdala. This peak did not show important between-study heterogeneity ($I^2 = 32\%$), but there was funnel plot asymmetry (meta-regression of Hedges' g by standard error p < 0.001) (for this and other funnel plots see S1 File).

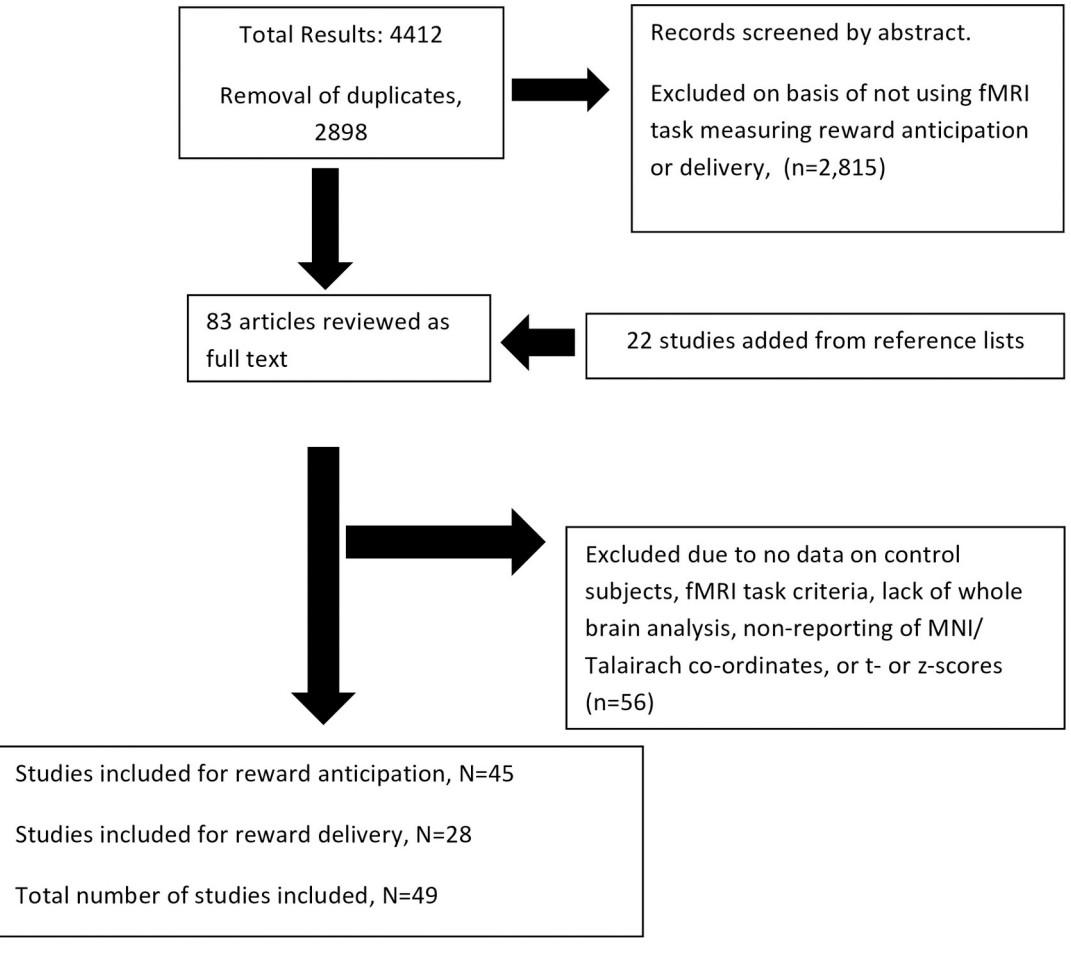

**Fig 1. Flow chart of studies considered for the meta-analysis.**

Another large cluster (3961 voxels, peak at MNI 0,6,34 with Hedges' g = 0.39) was located medially and involved predominantly the bilateral supplementary motor area and the middle cingulate cortex. It also extended to reach the superior frontal gyrus bilaterally. This peak did not show between-study heterogeneity ($I^2$ = 5%), and in this case there were no indications of publication bias (p = 0.18).

Among the smaller clusters, one (667 voxels, peak at MNI 44,-20,48 with Hedges' g = 0.29) was located in the right precentral cortex. This peak did not show between-study heterogeneity ($I^2$ = 5%) or indications of publication bias (p = 0.44). Another (633 voxels, peak at MNI -20,-96,-16 with Hedges' g = 0.28) was located in left inferior occipital cortex. Again, this peak did not show between-study heterogeneity ($I^2$ = 15%) or indications of publication bias (p = 0.26). Another (315 voxels, peak at MNI 38,-86,-8 with Hedges' g = 0.33) was located in right inferior occipital cortex. Between-study heterogeneity was low ($I^2$ = 24%) but there was funnel plot asymmetry (p = 0.04). A further cluster (289 voxels, peak at MNI -34,-6,56 with Hedges' g = 0.28) was located in the left precentral cortex. Again, the peak did not show between-study heterogeneity ($I^2$ = 17%) but there was funnel plot asymmetry (p = 0.04).

Three clusters were seen in the cerebellum. One (623 voxels, peak at MNI 8,-62,-14 with Hedges' g = 0.3) was located mostly on the right in lobules IV/V and VI. Between-study heterogeneity was low ($I^2$ = 29%) but there was significant funnel plot asymmetry (p = 0.03).

**Table 1. Included studies.**

| Study | N | Mean age | Country | MID type task | Task prelearnt | Reported anticipation data | Reported delivery data |
|---|---|---|---|---|---|---|---|
| Knutson et al [25] | 9 | 26.4 | USA | √ | √ | √ | √ |
| Kirsch et al [42] | 27 | 23.3 | Germany | √ | *x* | √ | *x* |
| Ramnani et al [43] | 8 | Not stated | UK | √ | √ | √ | *x* |
| Ernst et al [44] | 17 | 28.9 | USA | *x* | √ | √ | *x* |
| Wittman et al [45] | 16 | 22.9 | Germany | √ | √ | √ | *x* |
| Ernst et al [46] | 14 | range 20–40 | USA | *x* | √ | *x* | √ |
| Pessiglione et al [47] | 14 | range 19–37 | UK | *x* | *x* | √ | √ |
| Samanez-Larkin et al [48] | 12 | range 19–27 | USA | √ | √ | √ | √ |
| Bjork et al [49] | 20 | 34 | USA | √ | √ | √ | √ |
| Koch et al [50] | 28 | 24.6 | Germany | *x* | *x* | √ | √ |
| Knutson et al (2008) | 12 | 28.7 | USA | √ | √ | √ | √ |
| Dreher et al [51] | 20 | 25 | USA | *x* | √ | √ | √ |
| Schott et al [52] | 11 | 22.8 | Germany | √ | √ | *x* | √ |
| Cooper et al [53] | 12 | range 18–28 | USA | √ | √ | √ | *x* |
| Croxson et al [54] | 16 | range 19–27 | UK | √ | √ | √ | *x* |
| Roiser et al [55] | 19 | 27 | UK | *x* | *x* | √ | √ |
| Simon et al [56] | 24 | 24.8 | Germany | √ | √ | √ | √ |
| Koch et al [57] | 20 | 29.7 | Germany | *x* | *x* | √ | *x* |
| Jung et al [58] | 20 | 24.7 | South Korea | √ | √ | √ | √ |
| Kim et al [59] | 18 | 26.5 | USA/Ireland/Korea | √ | *?* | *x* | √ |
| Ivanov et al [60] | 16 | 30.6 | USA | √ | √ | √ | √ |
| Balodis et al [61] | 14 | 37.1 | USA | √ | √ | √ | *x* |
| Yau et al [62] | 20 | 20.1 | USA | √ | √ | √ | *x* |
| Filbey et al [63] | 27 | 30.3 | USA | √ | √ | √ | *x* |
| Rose et al [64] | 28 | 30.1 | USA | √ | √ | √ | √ |
| Saji et al [65] | 18 | 29.6 | Japan | √ | √ | √ | *x* |
| Kaufman et al [66] | 19 | 34.9 | Germany | √ | √ | √ | *x* |
| Costumero et al [67] | 44 | 23.4 | Spain | √ | √ | √ | *x* |
| Eppinger et al [68] | 28 | 49.4 | USA | *x* | √ | *x* | √ |
| Bustamente et al [69] | 18 | 37.44 | Spain | √ | √ | √ | √ |
| Damiano et al [70] | 31 | 23.58 | USA | √ | *?* | √ | √ |
| Funayama et al [71] | 20 | 29.9 | Japan | √ | √ | √ | *x* |
| Maresh et al [72] | 84 | 24.56 | USA | √ | √ | √ | √ |
| Weiland et al [73] | 12 | 30.9 | USA | √ | √ | √ | *x* |
| Wu et al [74] | 52 | 50 | USA | √ | √ | √ | √ |
| Behan et al [75] | 28 | 23 | Ireland | √ | √ | √ | *x* |
| Smieskova et al [76] | 19 | 26.4 | Switzerland | √ | *x* | √ | *x* |
| Mucci et al [77] | 22 | 31.9 | Italy | √ | √ | √ | √ |
| Kirk et al (2015) | 44 | 36.5 | USA | √ | *?* | √ | √ |
| Romanczuk-Seiferth et al [78] | 17 | 37.41 | Germany | √ | √ | √ | √ |
| Ubl et al [79] | 28 | 43.96 | Germany | √ | *?* | √ | √ |
| Carl et al [80] | 20 | 31.1 | USA | √ | √ | √ | √ |
| Yan et al [81] | 22 | 19 | China | √ | *?* | √ | √ |
| Herbort et al [82] | 23 | 25.78 | Germany | √ | √ | √ | *x* |
| Kollmann et al (2017) | 41 | 39.03 | Germany | √ | *X* | √ | *x* |
| He et al (2019) | 20 | 19.45 | China | √ | √ | √ | √ |
| Michielse et al (2019) | 40 | 21.9 | Holland/Belgium/UK | √ | √ | √ | *x* |

*(Continued)*

**Table 1.** (Continued)

| Study | N | Mean age | Country | MID type task | Task prelearnt | Reported anticipation data | Reported delivery data |
|---|---|---|---|---|---|---|---|
| Kim et al (2019) | 18 | 46 | USA | √ | √ | √ | *x* |
| Dhingra et al (2020) | 54 | 40 | USA | √ | ? | √ | √ |

Another (142 voxels, peak at MNI -28,-68,-32 with Hedges' g = 0.25) was located on the left (crus I and lobule IV). This peak did not show between-study heterogeneity ($I^2$ = 7%) or indications of funnel plot asymmetry (p = 0.58). The third (196 voxels, peak at MNI -24,-46,-30 with Hedges' g = 0.26) was located in right cerebellum (lobule VI). Again, the peak did not show between-study heterogeneity ($I^2$ = 1%) or indications of funnel plot asymmetry (p = 0.49).

Repeating the analysis at a lower threshold (FWER < 0.05) with clusters of at least 10 voxels led to a broadly similar pattern (see S1 File). The main difference was the appearance of additional small clusters (10–57 voxels) in the left superior temporal cortex, the left middle frontal cortex, the left insula and parts of the parietal and occipital cortex.

### Reward delivery

The analysis here resulted in three clusters of statistically significant voxels. The findings are shown in Fig 3 and Table 2. The largest (2830 voxels, peak at 0,46,6 with Hedges' g = 0.44) was in the anterior cingulate/medial frontal cortex bilaterally, extending to the gyrus rectus. Its

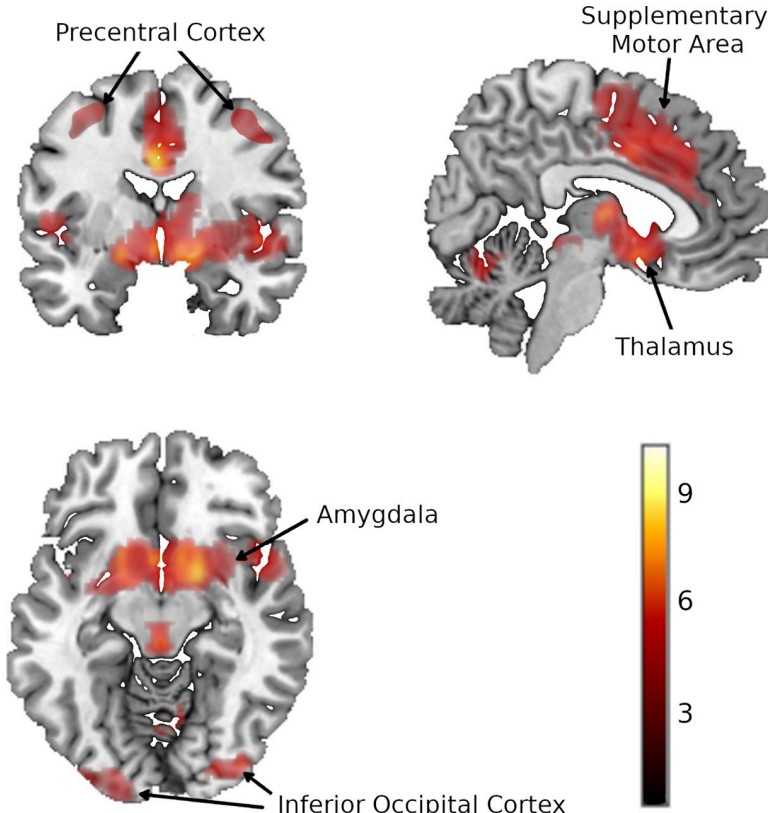

**Fig 2. Activations associated with monetary reward anticipation in 45 studies.** Depth of colour is proportional to the Z-value.

**Table 2. Results of the reward anticipation and delivery meta-analyses.**

| | Peak | | | | Cluster | |
|---|---|---|---|---|---|---|
| | MNI | Hedges' *g* | Z-value | FWER | Voxels | Breakdown |
| *Reward anticipation* | | | | | | |
| Striatum | 16,6,-12 | 0.42 | 8.9 | <0.001 | 5644 | B striatum (1648)<br>B insula (682)<br>B superior temporal cortex (311)<br>B inferior frontal cortex (204)<br>B olfactory cortex (197)<br>B thalamus (159)<br>B amygdala (49)<br>B gyrus rectus (34) |
| Medial frontal cortex | 0,6,34 | 0.39 | 10.4 | <0.001 | 3961 | B middle cingulate cortex (1882)<br>B supplementary motor area (1472)<br>B superior frontal gyrus (265) |
| Right precentral cortex | 44,-20,48 | 0.29 | 7 | <0.001 | 667 | R precentral cortex (416)<br>R postcentral cortex (106)<br>R middle frontal cortex (55) |
| Left inferior occipital cortex | -20,-96,-16 | 0.28 | 6.2 | <0.001 | 633 | L inferior occipital cortex (387)<br>L lingual (75) |
| Cerebellum | 8,-62,-14 | 0.3 | 6.6 | <0.001 | 623 | B cerebellum, lobule VI (360)<br>B cerebellum, lobule IV/V (73) |
| Right inferior occipital cortex | 38,-86,-8 | 0.33 | 7.1 | <0.001 | 315 | R inferior occipital cortex (200)<br>R lingual (26) |
| Left precentral cortex | -34,-6,56 | 0.28 | 6. | <0.001 | 289 | L precentral cortex (213)<br>L postcentral cortex (71) |
| Right cerebellum | 24,-46,-30 | 0.26 | 6.4 | <0.001 | 196 | R cerebellum, lobule VI (85)<br>R cerebellum, lobule IV/V (28) |
| Left cerebellum | -28,-68,-32 | 0.25 | 5.6 | <0.001 | 142 | L cerebellum, crus I (83)<br>L cerebellum, lobule VI (52) |
| *Reward delivery* | | | | | | |
| Medial frontal cortex | 0,46,6 | 0.44 | 8.79 | <0.001 | 2830 | B anterior cingulate cortex (1286)<br>B medial frontal gyrus (1012)<br>B gyrus rectus (257) |
| Posterior cingulate cortex | 0,-26,34 | 0.37 | 7.05 | <0.001 | 1140 | B posterior cingulate cortex (225)<br>B median cingulate cortex (522) |
| Striatum | 10,14,-6 | 0.35 | 6.89 | <0.001 | 225 | R striatum (143) |

B: bilateral. L: left. R: right. FWER: familywise error rate of the peak, derived from the distribution of the maximum z-statistic in a permutation test. MNI: coordinates of the peak in Montreal Neurological Institute space.

peak did not show relevant between-study heterogeneity ($I^2$ = 11%), and only a trend towards funnel plot asymmetry (p = 0.05) (see Fig 3).

A second, smaller cluster (1140 voxels, peak at 0,-26,34 with Hedges' g = 0.37) involved the bilateral posterior cingulate cortex. Its peak did not show significant between-study heterogeneity ($I^2$ = 14%) and only a trend towards funnel plot asymmetry (p = 0.06) (for funnel plots see S1 File).

The third cluster (225 voxels) was located in the right ventral striatum. This showed neither heterogeneity ($I^2$ = 6%) nor indications of publication bias (p = 0.25) (see Fig 2B).

Repeating the analysis at a lower threshold (FWER < 0.05) with clusters of at least 10 led to the appearance of a further small cluster (14 voxels) in the right angular gyrus (see S1 File).

## Analysis of moderator variables

**Reward anticipation.** Using a threshold of p = 0.01, and considering clusters with extension of 100 voxels or more, no clusters of correlation emerged with money per trial, percent

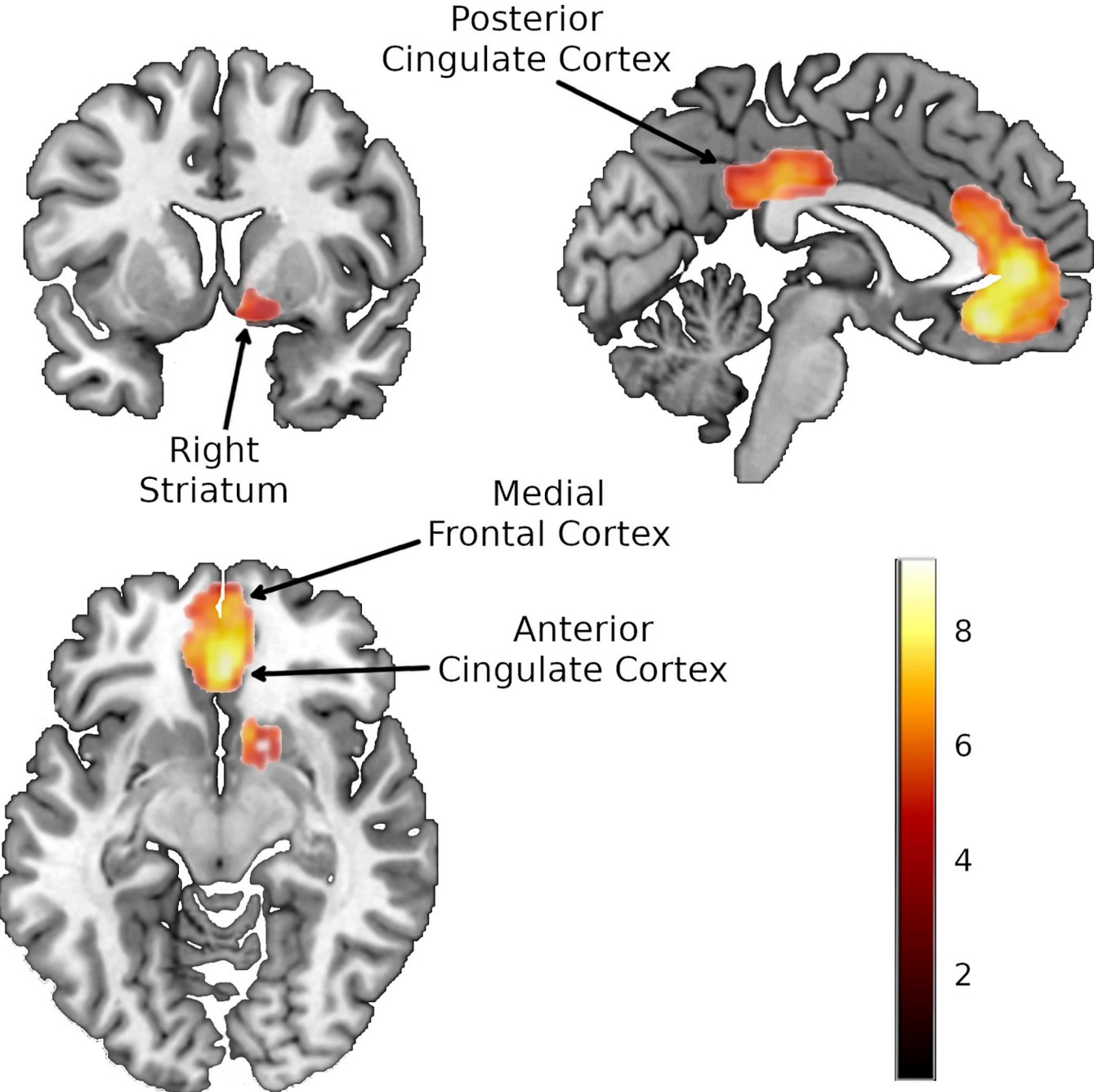

**Fig 3. Activations associated with monetary reward delivery in 28 studies.** Depth of colour is proportional to Z-value.

trials rewarded, percent male or smoothing. A cluster of 133 voxels in the left lateral middle frontal gyrus (133 voxels, BA 46, peak activation at -28,44,26, Z = 2.44, p = 0.0001) showed a positive correlation with the use of a pre-learnt task (i.e. stronger activation when the task was pre-learnt); this region was not present in the main analysis.

**Reward delivery.** No clusters of correlation were seen with money per trial or percent male. As in the meta-analysis of reward anticipation, a cluster of significant positive association emerged for whether the task was pre-learnt (i.e. stronger activation when the task was

pre-learnt). The cluster was in the medial prefrontal cortex (383 voxels, peak at -6,36,-14, BA 11, Z = 1.86, p = 0.001); this cluster overlapped with the medial frontal cortex cluster in the main analysis. A cluster of 183 voxels in the left lateral precentral gyrus (183 voxels, BA 6, peak activation at, peak at -48,2,34, Z = 2.5, p = 0.001) showed a positive correlation with smoothing (i.e. images with higher smoothing showed stronger activation); this region was not present in the main analysis.

## Discussion

This meta-analysis pooled data from fMRI studies using monetary reward in which there was an explicit cue:reward contingency. It found clear evidence of striatal activation, which was largely localized to the ventral striatal sector and was unilateral in the case of reward delivery. Other subcortical activations were minimal, but cortical activations were prominent–in the mid-cingulate gyrus/supplementary motor area in the case of anticipation, and in the anterior and posterior cingulate cortex the case of delivery. Only reward anticipation was associated with activations outside the medial cortical surface, which were located in the bilateral insula bilaterally and and the precentral cortex.

Striatal activations in our meta-analysis were strikingly restricted to the ventral striatal sector both for reward anticipation and reward delivery (in the case of anticipation, inspection of the peaks contributing to an apparent dorsal extension of the cluster revealed that the majority were actually in the thalamus or the insula). This finding stands in contrast to the findings of previous meta-analyses where a broader pattern of activation involving both the dorsal and ventral striatum has typically been found [27–31]. This was also the case in the two meta-analyses that were most similar to ours, in that they pooled data from tasks using monetary reward, specifically the MID task. Thus, Oldham et al [27] found relatively large bilateral clusters of activation involving both the ventral and dorsal striatum during reward anticipation although the localization was more exclusively ventral during delivery. Wilson et al [28] also found both dorsal and ventral striatal activation during anticipation (these authors did not examine delivery). Why our findings and those of previous meta-analyses differ in this respect is uncertain, though it does not seem to be a reflection of the numbers of studies included, since Oldham et al [27] included a similar number of studies to our meta-analysis and the number of studies in Wilson et al [28] was smaller at 15.

Our finding of activation in predominantly ventral striatal regions is consistent with the widely held belief that the ventral striatum, as opposed to the dorsal striatum, has a particular role in reward processing. This view rests partly on anatomical considerations–the ventral striatum forms part of a 'limbic' cortico-basal ganglia-cortical circuit which receives its cortical input from the prefrontal cortex, hippocampus and amygdala [14], and partly on findings from single cell recording studies in animals, which have found that reward sensitive neurons are twice as frequent in the ventral than in the dorsal striatum during reward delivery [83]. Also relevant is Schultz et al's [4] study of midbrain dopamine neurons that signal reward prediction error; this found a greater proportion of such neurons ('which occasionally reach statistical significance') in the ventral tegmental area and medial substantia nigra; these neurons project to the ventral striatum, albeit among other regions [5].

Areas in the medial cerebral cortex were prominent among cortical regions where we found reward-associated activations. It is also noteworthy that the areas here were different for anticipation and delivery–anticipation was associated with activation in a single bilateral cluster involving the mid-cingulate cortex and the supplementary motor area, whereas delivery activated two clusters, one in the ventromedial frontal cortex and another in the posterior cingulate cortex. These findings are similar to those of other meta-analyses. Thus, Diekhof et al

[31], Oldham et al, [27] and Wilson et al [28] all found a cluster (or in one case two clusters) of anticipation-associated activation in the mid-cingulate cortex/ supplementary motor area. Diekhof et al [31] and Oldham et al [27] found ventromedial frontal activation in association with reward delivery, with Oldham et al [27] additionally finding a cluster in the posterior cingulate cortex. Finally, Bartra et al [30] found two not-dissimilarly placed clusters to ours associated with delivery in their large meta-analysis using different kinds of reward tasks.

Why the mid-cingulate cortex and supplementary motor area might be involved in reward processing, specifically reward anticipation, is unclear. Animal studies have reported reward sensitive neurons in the the anterior cingulate cortex in rats [84], and the dorsal anterior cingulate cortex in monkeys [85], though in both cases this was at the time of delivery rather than during anticipation. In man, the mid-cingulate cortex and part of the pre-supplementary motor area are implicated in executive function or cognitive control. In particular, this region is activated by tasks requiring inhibition of prepotent responses [86], and has been argued to undertake specifically the 'evaluative' as opposed to 'regulative' cognitive control functions, monitoring the execution of plans generated to achieve task goals and signaling when adjustments are necessary [87–89]. These findings do not establish a link with reward processing, but it is interesting to note that in a review of the evidence for the function of different regions of the medial frontal cortex, Amodio and Frith [90] argued that its posterior and rostral zone (ie much the same area) acts to guide behaviour in terms of monitoring of the value of possible future actions. Clearly, such a function could easily encompass reward prediction.

In contrast, the ventromedial frontal cortex, one of the two medial cortex clusters activated by reward delivery in our and other meta-analyses, has a long tradition of being involved in reward processing. In particular, animal studies have identified the orbitofrontal cortex as an important region for multiple aspects of reward processing [7, 9]; this lies adjacent to the ventromedial frontal cortex and the two regions are often considered to form a single entity on anatomical grounds [91]. It is not clear why activation was restricted to the ventromedial frontal cortex in our meta-analysis, whereas others have found activation in both regions [27, 29, 31], but it could be related to the fact that the two regions have different patterns of anatomical connections [91], or alternatively to signal dropout in the orbitofrontal cortex.

In our meta-anlaysis, as in two others [27, 30], reward deliverly also activated the posterior cingulate cortex. The functions of this region have been reviewed by Leech and Sharp [92] and include attention, autobiographical memory and conscious awareness, but not, it should be noted, reward. The other notable feature of this region is that it, along with the medial frontal cortex, forms one of the two midline 'hubs' of the default mode network [93], a set of brain regions that de-activate during performance of a wide range of attention demanding tasks. The default mode network is also known to activate in response to some tasks, whose common feature appears to be the involvement of internally oriented, non-stimulus directed thought– examples include recall of autobiographical memories, imagining the future and theory of mind processes [93, 94]. An attempt has recently been made to integrate the role of the default mode network with reward processing on theoretical grounds [95]. However, it seems unlikely that reward is simply a further mental process that activates the default mode network: Wilson et al [28] argued for a pattern of both default mode activations and de-activations in their meta-analyis of reward anticipation, and Martins et al [96] found de-activations in the posterior cingulate cortex, angular gyrus, inferior parietal lobe and medial prefrontal cortex in a meta-analysis of studies of social rewards.

The main non-medial cortical region where we found activations was the insula, which was bilaterally activated by reward anticipation, but not by delivery. Diekhof et al [31] and Oldham et al [27] likewise found bilateral anterior insula activation in association with anticipation but not during delivery. Liu et al [29] found insula activation in their combined analysis of

anticipation and delivery, and Bartra et al [30] found it during their meta-analysis of reward delivery.

Although a brain region homologous with the insula exists in primates, the wide range of functions that it has been identified with do not include reward processing [97, 98]. Nevertheless, independent support for an role of the insula in reward comes from human studies. Seeley et al [99] applied independent component analyses (ICA) to resting-state fMRI data in 14 healthy subjects and found evidence not only for two previously well-characterized brain networks, the executive or cognitive control network and the default mode network, but also for a network involving the anterior insula, the dorsal anterior cingulate cortex, and the amygdala, substantia nigra/ventral tegmental area and thalamus. They [99] and subsequently Menon and Uddin [100] used the term 'salience network' to describe this set of brain regions, and hypothesized that it functioned to identify the most relevant among competing internal and extrapersonal stimuli for current behaviour. While they considered that these functions explicitly included reward, their conception of salience was broader than this, as 'a higher-order system for competitive, context-specific, stimulus selection and for focusing the 'spotlight of attention' and enhancing access to resources needed for goal-directed behavior'.

Two other cortical regions we found to be activated by reward anticipation have also been found in other meta-analyses. One was the precentral cortex, also found by Oldham et al [27], and the occipital cortex, found by Diekhof et al [31]. As noted in the Introduction, the former of these regions, though not the latter, has been found to be sensitive to reward processing in animal studies, but beyond this, the significance of these findings has to be regarded as obscure. Also unclear is the interpretation of our finding of activations in various subregions of the cerebellum. This finding is conspicuous by its absence in most of the meta-analyses cited above; only Wilson et al [28] found it in their meta-analysis of 15 MID studies which utilized information from individual group maps rather than peak co-ordinate. Recently, however, a direct cerebellar projection to the ventral tegmental area in mice has been discovered [101], which the authors also implicated in reward processing at the behavioural level.

There was no clear evidence of between-study heterogeneity in any of our findings ($I^2$ <50% for all clusters). This suggests that combining studies using the MID task with those using other monetary tasks, as we did, is a viable meta-analytic strategy. It might also imply that the methodological differences among studies (e.g., in the acquisition and processing of data) did not have important effects. On the other hand, there was evidence of publication bias affecting around half of the peaks of the clusters that emerged in our two meta-analyses. It is possible that this reflects the existence of significant numbers of studies that have gone unpublished because they found no activations associated with reward processing or in which the pattern was out of step with the existing literature. Possibly relevant to this explanation is that that the highest level of heterogeneity ($p<0.001$) found was in the large cluster centred on the ventral striatum in the anticipation meta-analysis, arguably the paradigmatic finding in the field.

Analysis of moderator variables was unrevealing, though in studies where the task was prelearnt before scanning, activations tended to be stronger in both reward anticipation and delivery. This seems an intuitive result, at least for reward anticipation, given that having to establish a cue: reward contingency while being scanned will inevitably delay and/or reduce the opportunities for a reward predictive cue to produce activations. Against such an interpretation, these associations were only seen in regions that did not emerge in the main meta-analyses. We found only minimal support for Sacchet and Knutson's [39] suggestion that use of smaller spatial smoothing kernels could exert an influence on the pattern of activations.

In conclusion, this meta-analysis finds that monetary reward anticipation and delivery are associated with partially dissociated pattern of brain activations. Anticipation activates a

network of regions whose core could be considered to consist of the ventral striatum, the mid-cingulate gyrus/supplementary motor area and the bilateral insula. Reward delivery also activates the ventral striatum, though to a smaller extent, and is otherwise associated with activations that are restricted to the anterior and posterior cingulate cortex. Our findings are the first to relate human reward processing specifically to the ventral striatum, something that is widely accepted from the animal literature. On the other hand, two of the cortical regions we found to be activated, the mid-cingulate cortex/supplementary motor area by anticipation and the posterior cingulate cortex by delivery, do not as yet have clearly defined reward-related roles.

## Supporting information

**S1 Checklist. Jauhar et al PRISMA checklist.**
(DOCX)

**S1 File. List of excluded studies, funnel plots, meta-analysis results at a lower threshold of (FWER < 0.05).**
(DOCX)

## Author Contributions

**Conceptualization:** S. Jauhar, P. J. McKenna, J. Radua.

**Data curation:** S. Jauhar, L. Fortea, A. Solanes, A. Albajes-Eizagirre, P. J. McKenna, J. Radua.

**Formal analysis:** L. Fortea, A. Solanes, A. Albajes-Eizagirre, J. Radua.

**Funding acquisition:** J. Radua.

**Investigation:** S. Jauhar, L. Fortea, A. Solanes, A. Albajes-Eizagirre.

**Methodology:** S. Jauhar, P. J. McKenna, J. Radua.

**Project administration:** J. Radua.

**Resources:** A. Albajes-Eizagirre.

**Software:** A. Albajes-Eizagirre.

**Supervision:** P. J. McKenna, J. Radua.

**Validation:** L. Fortea.

**Writing – original draft:** P. J. McKenna, J. Radua.

**Writing – review & editing:** S. Jauhar, L. Fortea, P. J. McKenna.

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
