## [Decision Letter · Decision Letter 0]

22 Mar 2021

PONE-D-20-35918

Brain activations associated with anticipation and delivery of monetary reward: a systematic review and meta-analysis of fMRI studies

PLOS ONE

Dear Dr. McKenna,

Thank you for submitting your manuscript to PLOS ONE. After careful consideration, we feel that it has merit but does not fully meet PLOS ONE’s publication criteria as it currently stands. Therefore, we invite you to submit a revised version of the manuscript that addresses the points raised during the review process.

Please address issues raised by two reviewers.

We look forward to receiving your revised manuscript.

Kind regards,

Wi Hoon Jung, PhD

Academic Editor

PLOS ONE

Journal Requirements:

Reviewers' comments:

Reviewer's Responses to Questions

**Comments to the Author**

1. Is the manuscript technically sound, and do the data support the conclusions?

Reviewer #1: Yes

Reviewer #2: Partly

2. Has the statistical analysis been performed appropriately and rigorously? 

Reviewer #1: Yes

Reviewer #2: Yes

3. Have the authors made all data underlying the findings in their manuscript fully available?

Reviewer #1: Yes

Reviewer #2: No

4. Is the manuscript presented in an intelligible fashion and written in standard English?

Reviewer #1: Yes

Reviewer #2: No

5. Review Comments to the Author

Reviewer #1: In this work, Jauhar et al. performed a SDM-PSI meta-analysis of fMRI studies investigating the anticipation and receipt of financial rewards, as assessed using the Monetary Incentive Delay Task. The authors found that both anticipation and receipt of financial rewards engage the ventral striatum but are associated with different patterns of cortical BOLD activity.

I commend the authors for putting together a clear and well written manuscript. However, my main concern relates to the novelty or impact these findings might have for the broad audience. A number of meta-analyses have been conducted on this topic (including a recent one on data from the anticipation phase of the MID task using SDM). Hence, from the current version of the manuscript, it is not clear what we should learn from this manuscript that might advance our understanding of the brain processing of financial rewards. Also, while the authors base their approach on SDM, which allows to include statistical maps to increase sensitivity and overcome problems related to statistical power, no attempt was made to retrieve and include statistical maps - which is not only a missed opportunity but could add to their findings on receipt (which as far as i am concerned have never been meta-analyzed using SDM and could represent the biggest conceptual advance of this manuscript).

Minor points:

Abstract:

please describe the moderator effect in further detail

Introduction:

The authors outline well their choices to focus only on monetary tasks, excluding those which did not include a neutral condition; however, it is not clear from the Introduction why this is important to move the field forward. In particular, the conceptual advance beyond Wilson et al. should be explained in further detail, which is to my knowledge, the closest work on the same topic and using the same approach. The authors are correct in stating that mixing different types of primary rewards might bring heterogeneity; however, on the other side one can also argue that the use of a range of stimuli is more representative of the richness of real-life and might enhance our confidence that the reported results are not confined to specific lab parameters.

Can the authors expand the introduction to explain the advantages of SDM-PSI, when compared to other meta-analytic models? This might sound trivial to the authors but could help the reader naive to the method to establish parallels with other coordinate-based approaches on the same topic.

As a general comment, i discourage the authors to use the terms "activations" or "deactivations", which although widely used in fMRI research can more accurately be described as increases and decreases in BOLD signal, respectively.

Methods:

The authors provide a vague rationale to refer to blobs instead of clusters - i find the first a bit too simplistic and invite the authors to reformulate, so the reader less familiar with the method but more familiar with cluster-level analysis in fMRI can follow the results more intuitively. I note though that in the discussion the authors then refer to clusters. Please keep the terminology consistent so the text is easier to follow and do not raise unnecessary questions.

Can the authors provide a justification for the statistical threshold they selected to identify the significant "blobs"? I wonder whether their option might have been too stringent and can explain differences in cortical regions from Wilson et al.

Discussion:

Might the absence of findings in the orbitofrontal cortex reflect poor coverage of this region due to signal drop out in a considerable number of studies? This is an instance where having gathered statistical maps would have helped to appraise the results further.

Although the main findings of this meta-analysis are presented, it does not provide convincing evidence that these findings can be useful for researchers or clinicians. That is, the importance of these meta-analysis findings is not provided strongly in the discussion section

Reviewer #2: The authors aimed to explore neural mechanisms of reward anticipation and delivery by a fMRI meta-analysis and compare similarities and differences between these functional brain activations. They included previous literatures of fMRI studies that included healthy volunteers and used monetary reward tasks with reward cues. They analyzed the SDM and I-squared heterogeneity test for the meta-analysis. The authors found functional responses to reward anticipation in striatum, SMA, insula, etc., while ventral striatum, anterior and posterior cingulate gyri are involved in reward delivery. Although these findings were in accordance with the previous literatures, the rationale of the current study seems unclear. Also, the findings of the study seem not to be sufficiently interpreted.

1. The rationale of the current study is not clearly described in the abstract and introduction. Please clarify which unanswered questions or controversial issues this study aimed to address.

- in the last paragraph of the introduction, authors introduced their research aims as follows: “a) restrict the analysis to studies using monetary reward … and b) to broaden the inclusion criteria from the MID task to any study that utilized an overt monetary cue-reward contingency”. However, it was not described how the authors developed this aim from previous literatures. I would like to ask them to describe reasons for the restriction of their scope to the cue-based monetary reward process. For example, they may want to clarify a) what controversies and issues were noticed from previous studies and b) how the current study design can improve these problems.

- Previously, there were studies that had similar aims and study design. For example, Oldham et al. (2018) have already determined similarities and differences of neural activations during reward anticipation and delivery processes. Please describe how the current study advanced findings of this previous study.

2. In many parts of the discussion section, the authors seem to restate their neuroimaging results and mention if these findings were consistent or inconsistent to the previous findings. There is lack of explanations about what the differences would mean and why these controversies were resulted in.

3. In the discussion section, authors mentioned “Our meta-analysis is therefore provides the clearest evidence to date for reward processing in humans being associated particularly with ventral striatal activation ...”. Because it seems that there was no supporting evidence for this argument, they should provide evidence (e.g., methodological improvement, superiority in study design) for this argument.

Please also see minor comments blow.

4. Please use clear label names for brain regions in the abstract and results.

5. Regarding the study exclusion criteria, please provide a) the age ranges that the authors considered as elderly and adolescent samples and b) reasons for excluding “… studies where participants did not actually receive money at the end of the study ...” in the methods section.

6. Correcting typographical and punctuation errors is needed.

7. For the figures 1 & 2, providing the statistical maps would be more informative than the current binary mask images.

8. Please explain which brain atlas system was used to label brain regions in the results section.

6. PLOS authors have the option to publish the peer review history of their article (what does this mean?). If published, this will include your full peer review and any attached files.

Reviewer #1: No

Reviewer #2: No

---

## [Author Response · Author response to Decision Letter 0]

20 May 2021

Dear Dr/Prof Wi Hoon Jung

Thank you for considering a revised version of this ms. We have made changes addressing almost all the issues raised, the only exception being that of Reviewer #1 who suggested that we carry out what would have been essentially a new and different meta-analysis based on obtaining statistical maps from study authors. We explain why we did not feel this was appropriate or feasible in the detailed responses below.

We hope the paper is now acceptable. We would of course be willing to make further changes if necessary.

Yours sincerely 

Peter McKenna, corresponding author.

Reviewer #1: In this work, Jauhar et al. performed a SDM-PSI meta-analysis of fMRI studies investigating the anticipation and receipt of financial rewards, as assessed using the Monetary Incentive Delay Task. The authors found that both anticipation and receipt of financial rewards engage the ventral striatum but are associated with different patterns of cortical BOLD activity.

I commend the authors for putting together a clear and well written manuscript. However, my main concern relates to the novelty or impact these findings might have for the broad audience. A number of meta-analyses have been conducted on this topic (including a recent one on data from the anticipation phase of the MID task using SDM). Hence, from the current version of the manuscript, it is not clear what we should learn from this manuscript that might advance our understanding of the brain processing of financial rewards. 

- We have now expanded on this issue in the introduction. In particular, we provide more justification for a) why inclusion of studies using non-monetary rewards could be problematic (specifically, that it is not clear that tasks using social rewards will activate the same areas as more concrete rewards like money or food); and b) why we did not include gambling and delay discounting tasks (the lack of cue-reward contingency means that anticipation cannot be easily examined).

- We now also highlight an issue in the Introduction that we previously only mentioned in passing the methods. This is that, when meta-analyzing voxel-based studies it is important to exclude studies that did not employ whole brain coverage. This is something that as far as we can tell previous meta-analyses have not addressed.

Also, while the authors base their approach on SDM, which allows to include statistical maps to increase sensitivity and overcome problems related to statistical power, no attempt was made to retrieve and include statistical maps - which is not only a missed opportunity but could add to their findings on receipt (which as far as i am concerned have never been meta-analyzed using SDM and could represent the biggest conceptual advance of this manuscript).

- Obtaining statistical maps was not part of our research plan. Doing so at this late stage would require writing to around 50 authors, plus undertaking a complete re-analysis of the data. We also note that it is likely that the response rate of authors would be low – in Wilson et al’s 2018 meta-analysis that took this approach, maps were only obtained in response to 36 of 108 requests, and only 15 maps were finally used. We request that our approach be judged on its own merits.

Minor points:

Abstract:

please describe the moderator effect in further detail

- We now do this.

Introduction:

The authors outline well their choices to focus only on monetary tasks, excluding those which did not include a neutral condition; however, it is not clear from the Introduction why this is important to move the field forward. In particular, the conceptual advance beyond Wilson et al. should be explained in further detail, which is to my knowledge, the closest work on the same topic and using the same approach. The authors are correct in stating that mixing different types of primary rewards might bring heterogeneity; however, on the other side one can also argue that the use of a range of stimuli is more representative of the richness of real-life and might enhance our confidence that the reported results are not confined to specific lab parameters.

- For our choice to focus exclusively on monetary tasks, see comments above. 

- We see the approach taken in our meta-analysis as complementary to those of existing reward fMRI meta-analyses. Clearly, there are many different approaches that can be (and have been) taken to examine reward fMRI studies meta-analytically, and we feel ours is valid and answers some outstanding questions in the literature. 

Can the authors expand the introduction to explain the advantages of SDM-PSI, when compared to other meta-analytic models? This might sound trivial to the authors but could help the reader naive to the method to establish parallels with other coordinate-based approaches on the same topic.

- We now explain this, as follows: ‘…previous methods have relied on an unorthodox statistical test: they aim to find those voxels whether the convergence of findings is statistically significantly higher than in other voxels. Conversely, the statistical tests in SDM-PSI aim to find those voxels whether the BOLD response is statistically significant, ie, as in the standard statistical tests applied to original imaging data within SPM or FSL. The use of standard statistics has multiple additional benefits, such as taking the effect size into account (other previous methods do not), reporting standard estimates of between-study heterogeneity (e.g., the I2 statistic, interpreted as the percentage of variation unrelated to sampling error), and allowing standard tests for the detection of potential publication bias based on funnel plot asymmetry (for more details see [32]).’

As a general comment, i discourage the authors to use the terms "activations" or "deactivations", which although widely used in fMRI research can more accurately be described as increases and decreases in BOLD signal, respectively.

- Given that the term activations is pervasis in the literature, it would be unrealistic and also artificial to try and substitute ‘increases in BOLD signal’ throughout the paper. Instead, we have inserted a note on this point in the introduction saying, ‘throughout this article we use the term ‘activations’ for convenience, although strictly speaking what is referred to are increases blood flow/metabolism in PET and increases in BOLD signal in fMRI’.

Methods:

The authors provide a vague rationale to refer to blobs instead of clusters - i find the first a bit too simplistic and invite the authors to reformulate, so the reader less familiar with the method but more familiar with cluster-level analysis in fMRI can follow the results more intuitively. I note though that in the discussion the authors then refer to clusters. Please keep the terminology consistent so the text is easier to follow and do not raise unnecessary questions.

- We agree that the use of two terms was confusing. To avoid confusion, we now only refer to clusters throughout.

Can the authors provide a justification for the statistical threshold they selected to identify the significant "blobs"? I wonder whether their option might have been too stringent and can explain differences in cortical regions from Wilson et al.

- We used this threshold to ensure that the results were robust in the sense of being unlikely to be due to chance. We now additionally provide results at a less conservative threshold. Specifically, we now state: ‘In the text, we consider the most robust results (FWER < 0.01 in clusters of at least 100 voxels); for completeness, however, we also report results at a more liberal threshold (FWER < 0.05) with clusters of at least 10 voxels in the Supplementary Material.’ We summarize the findings from this additional analysis in the results section.

Discussion:

Might the absence of findings in the orbitofrontal cortex reflect poor coverage of this region due to signal drop out in a considerable number of studies? This is an instance where having gathered statistical maps would have helped to appraise the results further.

- We now note this possibility in the discussion.

Although the main findings of this meta-analysis are presented, it does not provide convincing evidence that these findings can be useful for researchers or clinicians. That is, the importance of these meta-analysis findings is not provided strongly in the discussion section.

- We have rewritten the final paragraph of the discussion to make the following points. (i) our meta-analysis finds that reward anticipation and delivery are associated with partially dissociated pattern of brain activations, the former activating the ventral striatum, the mid-cingulate gyrus/supplementary motor area and the bilateral insula, and the latter the ventral striatum and the anterior and posterior cingulate cortex; (ii) our meta-analysis is the first to specifically relate human reward processing to the ventral striatum, something that is widely accepted in the animal literature; and (iii) two of the cortical regions we found to be activated, the mid-cingulate cortex/supplementary motor area (anticipation) and the posterior cingulate cortex (delivery), do not as yet have clearly defined reward-related roles.

Reviewer #2: The authors aimed to explore neural mechanisms of reward anticipation and delivery by a fMRI meta-analysis and compare similarities and differences between these functional brain activations. They included previous literatures of fMRI studies that included healthy volunteers and used monetary reward tasks with reward cues. They analyzed the SDM and I-squared heterogeneity test for the meta-analysis. The authors found functional responses to reward anticipation in striatum, SMA, insula, etc., while ventral striatum, anterior and posterior cingulate gyri are involved in reward delivery. Although these findings were in accordance with the previous literatures, the rationale of the current study seems unclear. Also, the findings of the study seem not to be sufficiently interpreted.

1. The rationale of the current study is not clearly described in the abstract and introduction. Please clarify which unanswered questions or controversial issues this study aimed to address.

- See responses to Reviewer 1 above. We now describe the rationale for the meta-analysis more fully in the introduction. As noted in these responses, we see our meta-analysis mainly as complementing existing ones, though we address some significant outstanding issues (elaborated further below). 

in the last paragraph of the introduction, authors introduced their research aims as follows: “a) restrict the analysis to studies using monetary reward … and b) to broaden the inclusion criteria from the MID task to any study that utilized an overt monetary cue-reward contingency”. However, it was not described how the authors developed this aim from previous literatures. I would like to ask them to describe reasons for the restriction of their scope to the cue-based monetary reward process. For example, they may want to clarify a) what controversies and issues were noticed from previous studies and b) how the current study design can improve these problems.

- See response to Reviewer 1 above. We now give an explicit set of reasons for directing the meta-analysis to cue-based monetary reward studies beyond the monetary incentive delay task and excluding non-cue-based tasks.

- Our methodology of restricting the analysis to studies that employed full brain coverage is also relevant here.

Previously, there were studies that had similar aims and study design. For example, Oldham et al. (2018) have already determined similarities and differences of neural activations during reward anticipation and delivery processes. Please describe how the current study advanced findings of this previous study.

- There are two concrete ways in which we advance (or complement) the findings of Oldham et al. (i) They restricted their meta-analysis to studies using the monetary incentive delay task. (ii) As we now highlight, their and other meta-analyses did not exclude studies with only partial brain coverage. We now make these points explicitly in the Introduction.

2. In many parts of the discussion section, the authors seem to restate their neuroimaging results and mention if these findings were consistent or inconsistent to the previous findings. There is lack of explanations about what the differences would mean and why these controversies were resulted in.

- We have now extensively rewritten the discussion. On the one hand, we have cut down what we say about similarities and differences to other meta-analyses. On the other, we have increased what we say about the possible interpretation of the findings, making reference to animal findings and the potential function of brain areas that we (and the authors of other meta-analyses) found to be activated.

3. In the discussion section, authors mentioned “Our meta-analysis is therefore provides the clearest evidence to date for reward processing in humans being associated particularly with ventral striatal activation ...”. Because it seems that there was no supporting evidence for this argument, they should provide evidence (e.g., methodological improvement, superiority in study design) for this argument.

- We have double-checked and it does seem to be the case that our meta-analysis is the only one to date to find activations related to reward that are restricted to the ventral striatum. All other meta-analyses have to a greater or lesser extent implicated the dorsal striatum or sometimes the whole of the basal ganglia. 

- We have added text in the discussion on animal findings, which tend to implicate specifically the ventral in reward processing. 

- Based on this, we have now modified what we say in the final paragraph of the discussion to, ‘our meta-analysis the first to relate human reward processing specifically to the ventral striatum, something that is widely accepted in the animal literature’.

Please also see minor comments blow.

4. Please use clear label names for brain regions in the abstract and results.

- We have tried to be explicit about the activated brain regions throughout the paper.

5. Regarding the study exclusion criteria, please provide a) the age ranges that the authors considered as elderly and adolescent samples and b) reasons for excluding “… studies where participants did not actually receive money at the end of the study ...” in the methods section.

- We have now specified age ranges.

- With respect to the reason for excluding studies where the participants did not receive money, this was simply one of the a priori operational decisions that meta-analyses routinely have to make. Many such decisions are finely balanced, and it would certainly also have been possible to carry out the meta-analysis including studies where the participants received no money. The important thing is that such decisions are defensible, which we feel is obviously the case in this instance, and explicitly described, which we have done.

6. Correcting typographical and punctuation errors is needed.

- We have checked for errors.

7. For the figures 1 & 2, providing the statistical maps would be more informative than the current binary mask images.

- We have replaced the relevant figures with those showing statistical maps (the results are closely similar).

8. Please explain which brain atlas system was used to label brain regions in the results section.

- We are now state that all results are reported in MRI space and the brain regions are labelled according to the AAL atlas.

---

## [Decision Letter · Decision Letter 1]

14 Jul 2021

Brain activations associated with anticipation and delivery of monetary reward: a systematic review and meta-analysis of fMRI studies

PONE-D-20-35918R1

Dear Dr. McKenna,

We’re pleased to inform you that your manuscript has been judged scientifically suitable for publication and will be formally accepted for publication once it meets all outstanding technical requirements.

Kind regards,

Wi Hoon Jung, PhD

Academic Editor

PLOS ONE

Reviewers' comments:

Reviewer's Responses to Questions

**Comments to the Author**

1. If the authors have adequately addressed your comments raised in a previous round of review and you feel that this manuscript is now acceptable for publication, you may indicate that here to bypass the “Comments to the Author” section, enter your conflict of interest statement in the “Confidential to Editor” section, and submit your "Accept" recommendation.

Reviewer #1: All comments have been addressed

Reviewer #2: All comments have been addressed

2. Is the manuscript technically sound, and do the data support the conclusions?

Reviewer #1: Yes

Reviewer #2: Yes

3. Has the statistical analysis been performed appropriately and rigorously? 

Reviewer #1: Yes

Reviewer #2: Yes

4. Have the authors made all data underlying the findings in their manuscript fully available?

Reviewer #1: No

Reviewer #2: Yes

5. Is the manuscript presented in an intelligible fashion and written in standard English?

Reviewer #1: Yes

Reviewer #2: Yes

6. Review Comments to the Author

Reviewer #1: The authors have addressed all my comments. Hence, I recommend the manuscript for publication in its current form.

Reviewer #2: The authors addressed all comments raised in the previous round of review, improving their manuscript by 1) clearly describing rationale of this study and 2) complementing previous understanding about monetary reward-related brain activations. I recommend that this manuscript is acceptable for publication.

7. PLOS authors have the option to publish the peer review history of their article (what does this mean?). If published, this will include your full peer review and any attached files.

Reviewer #1: No

Reviewer #2: No

---

## [Editor Report · Acceptance letter]

27 Jul 2021

PONE-D-20-35918R1 

Brain activations associated with anticipation and delivery of monetary reward: a systematic review and meta-analysis of fMRI studies 

Dear Dr. McKenna:

I'm pleased to inform you that your manuscript has been deemed suitable for publication in PLOS ONE. Congratulations! Your manuscript is now with our production department. 

Kind regards, 

on behalf of

Dr. Wi Hoon Jung 

Academic Editor

PLOS ONE